# LARGE LANGUAGE MODELS OUTPERFORM STATE-OF-THE-ART METHODS ON MULTICLASS SENTIMENT POLARITY DETECTION

## ABSTRACT

Sentiment polarity detection remains a significant problem with applications such as opinion tracking and social network analysis. In this study, we evaluate whether contemporary open-weight large language models (LLMs) can rival or surpass specialized approaches on multiclass polarity detection while accounting for inference cost. We conduct a systematic zero-shot evaluation of 31 open-weight LLMs on two canonical five-class benchmarks, assessing Accuracy, Macro-Average Mean Absolute Error, and instances-per-second measures to quantify cost-performance trade-offs, identifying the best models according to the Pareto criterion. We found that several LLMs, without fine-tuning or elaborate prompting, outperform previous state-of-the-art results on a large dataset (SemEval) and approach a similar to state-of-the-art performance on a smaller benchmark (SST-5). Our Pareto frontier analysis highlights models that combine high accuracy with low inference costs, offering practical deployment choices for fine-grained sentiment polarity detection.

## 1 INTRODUCTION

Sentiment polarity detection is a fundamental task in natural language processing, enabling the extraction of opinions and attitudes from text in a format more easily interpreted in large-scale settings. It underpins diverse applications across industry and academia: for example, businesses use sentiment analysis for real-time brand monitoring and customer feedback analytics, while social scientists and public health researchers apply it to gauge public mood, political opinion, or population well-being. In fact, the field of sentiment analysis is often described by terms like "brand monitoring" or "buzz monitoring" to emphasize its practical role in measuring audience reactions. Despite decades of work, achieving nuanced and reliable sentiment understanding remains a core challenge in NLP.

Traditional sentiment polarity detection often reduces text to binary (positive/negative) or ternary (positive/neutral/negative) labels. However, many real-world scenarios demand finer-grained distinctions. A 5-class polarity framework (very negative, negative, neutral, positive, very positive) captures the intensity and subtlety that binary schemes miss, as it can distinguish, for example, between mildly dissatisfied and strongly disgruntled customers, or identify trends of gradually increasing public optimism. The adoption of a 5-class polarity scheme, or even more granular approaches, yields more nuanced and actionable insights into opinions than a coarse binary approach.

Recent advances in large language models (LLMs) have dramatically reshaped sentiment polarity detection tasks. Pretrained LLMs demonstrate strong natural language understanding, capturing subtle context and linguistic characteristics that previous models struggle with. For instance, ChatGPT-like models have been found to match fine-tuned classifiers on standard sentiment tasks, achieving performance comparable to BERT in binary sentiment analysis, as stated by Wang et al. (2024).

At the same time, an ecosystem of open-weight LLMs (models with publicly available parameters) has emerged. These models are particularly attractive, since they offer accessibility, privacy, and customizability. However, this growing variety of open LLMs brings a crucial cost-performance trade-off. Larger models generally yield higher accuracy, but demand more inference time and com-

putational costs. Choosing the "best" model for a given setting requires balancing predictive quality and efficiency. However, to the best of our knowledge, existing benchmarks have not explored this trade-off in the context of multiclass ordinal sentiment polarity detection.

To address this gap, we evaluate a set of publicly available models under a unified experimental framework that jointly measures performance and inference time. In our experiments, we use two canonical sentiment datasets, the Stanford Sentiment Treebank (5-way) and SemEval-2017 Task 4C, which contain compositional sentiments in movie reviews and tweets, respectively. By systematically varying model size and architecture across these benchmarks, our study explores how open LLMs balance performance and efficiency in fine-grained polarity detection. We demonstrate that a subset of the evaluated models surpasses the previous best results in the SemEval dataset, achieving a new state-of-the-art performance in the explored metrics.

This study is organized as follows. In the next section, we explore related publications, highlighting their main achievements, while showing the gaps our study seeks to fill. Section 3 introduces the methodology used to evaluate the models, detailing the criteria for selecting the assessed models, the datasets against which they are compared, and the metrics used to quantify their performance. Section 4 provides an analysis of the results and the (new) state-of-the-art performance. Concluding remarks are drawn in the last section. After the main text and references, we append the supplementary material, including the full list of evaluated models, the complete prompts provided to them, and their full performance report.

## 2 RELATED WORK

Large language models have been increasingly applied to sentiment analysis, though most studies rely on closed-weight systems. For instance, Wang et al. (2024) showed that ChatGPT performs comparably to fine-tuned BERT on binary sentiment classification, while Zhang et al. (2024) benchmarked LLMs across 13 sentiment tasks and found they excel in simple binary or ternary settings but struggle on complex, structured sentiment. Other works highlight the effectiveness of fine-tuned GPT models for fine-grained sentiment. Simmering & Huoviala (2023) demonstrated state-of-the-art results with GPT-3.5 on aspect-based sentiment, but noted this was achieved when increasing the number of parameters by a factor of 1000, which implied increased inference time. Similarly, Vamvourellis & Mehta (2025) evaluated GPT-4 variants for financial sentiment classification and observed that simple prompting strategies outperformed more elaborate reasoning setups. These studies confirm the viability of LLMs in sentiment analysis but focus on closed models and limited sentiment categories.

In contrast, several works evaluate open-weight models for sentiment tasks. Niimi (2024) used medium-sized open LLMs to annotate restaurant reviews and found that majority voting across multiple passes improved both accuracy and efficiency compared to a single large model. Nasution et al. (2025) compared 22 open LLMs against ChatGPT-4 on Indonesian tweets and showed that large open models like LLaMA3 70B and Gemma2 27B achieved over 90% of ChatGPT's macro-F1 while requiring significantly longer inference times. Ensemble approaches with open LLMs have also been explored, such as Salimian et al. (2025), who reported 6–15% gains in multiclass sentiment prediction from combining outputs of several models. More general benchmarks such as Bi et al. (2025) revealed that OpenAI's 20B GPT-OSS model outperformed a larger 120B variant on tasks like HumanEval and MMLU, underscoring the importance of considering both scale and efficiency. These works highlight the promise of open LLMs but rarely examine their performance in multiclass ordinal sentiment in a unified framework.

Beyond sentiment polarity as positive/negative, researchers have explored fine-grained and ordinal sentiment scales. Qin et al. (2024) proposed LAMPO, which reformulates multiclass sentiment as ordinal pairwise preferences, significantly improving performance on 5-class sentiment classification. Such methods emphasize the value of modeling ordered sentiment levels rather than treating them as flat categories. However, most existing LLM sentiment studies still remain limited to binary or ternary schemes. This leaves open questions about how these models respect the structure and granularity of sentiment scales.

Finally, several strands of work emphasize the importance of efficiency and cost-performance trade-offs in LLM evaluation. As stated, Simmering & Huoviala (2023) noted that GPT-3.5's sentiment

gains required vastly larger parameter counts, while Niimi (2024) demonstrated that iterative use of smaller open models could be faster and more accurate overall. Alassan et al. (2024) compared inference latencies across proprietary and open models, reporting that optimized smaller LLMs can achieve sub-30ms latencies with slight performance loss. Notably, Bi et al. (2025) also incorporated metrics like energy use and memory footprint into LLM benchmarking.

Building on these insights, our study analyzes the accuracy and efficiency trade-off in a comparative evaluation of several open-weight models in multiclass sentiment polarity detection tasks, while also demonstrating that they are able to outperform previous state-of-the-art results at a comparatively low inference cost.

## 3 METHODOLOGY

This section outlines the criteria used for choosing the evaluated models, the benchmark datasets, and the performance metrics.

### 3.1 MODELS

The selection of 31 open-weight Large Language Models (LLMs) for this study was guided by the principle of ensuring a diverse and representative sample of the contemporary LLM landscape. The selection was based on three criteria: the diversity of model lineages, varied training and attention mechanism optimizations, and a wide spectrum of parameter scales. By doing so, we can analyze the impact of these factors on both model performance and computational cost.

The selection includes models from major corporate labs (Meta, Google, Microsoft), specialized AI companies (Mistral AI, Cohere), and research-focused organizations (Allen Institute for AI). This diversity assesses whether performance trends are universal or specific to certain development paradigms. We included multiple models from the same lineage, such as successive versions of Llama, Gemma, Phi, Qwen, and Mistral, which enables a longitudinal analysis of how iterative improvements impact performance. Most selected models (Llama, Mistral, Gemma) are dense, activating all parameters for every token. However, we also included Mixture-of-Experts (MoE) models, such as qwen3 and gpt-oss. This helps explore the trade-off between a large total parameter count and a much smaller active parameter count during inference.

Additionally, the corpus contains models with various attention mechanism optimizations, such as the widely adopted Grouped-Query Attention (GQA), Mistral's Sliding Window Attention (SWA), and DeepSeek's novel Multi-Head Latent Attention (MLA), which enables a comprehensive evaluation of the impact of different strategies on sentiment polarity results.

Finally, the models were chosen to represent a broad spectrum of parameter scales. The parameter counts range from small-scale models (2B-8B) designed for on-device deployment to large-scale models (24B-32B).

The exhaustive list of all tested models can be found in the section C of the Appendix.

### 3.2 DATASETS

We assessed the performance of the models on two distinct and complementary dimensions of language understanding. **SemEval-2017 Task 4 Subtask C**, sourced from Rosenthal et al. (2017), offers topic-centric sentiment in informal discourse, while **Stanford Sentiment Treebank (SST-5)**, sourced from Socher et al. (2013), enables the analysis of compositional sentiment in structured text. These are two canonical NLP benchmarks that have been widely explored in the existing literature.

SemEval-2017 Task 4 Subtask C, whose instances were sourced from X (formerly Twitter), requires models to determine sentiment polarity in a given tweet, provided the context (the hashtag under which the tweet was posted). This task requires targeted, pragmatic reasoning, given that the text is characterized by brevity, non-standard grammar, slang, and other linguistic phenomena typical of social media. The core challenge lies in extracting the sentiment directed to a specific entity, separating it from other sentiments expressed in the text, a process that tests the effectiveness of an LLM's attention mechanisms in a zero-shot setting.

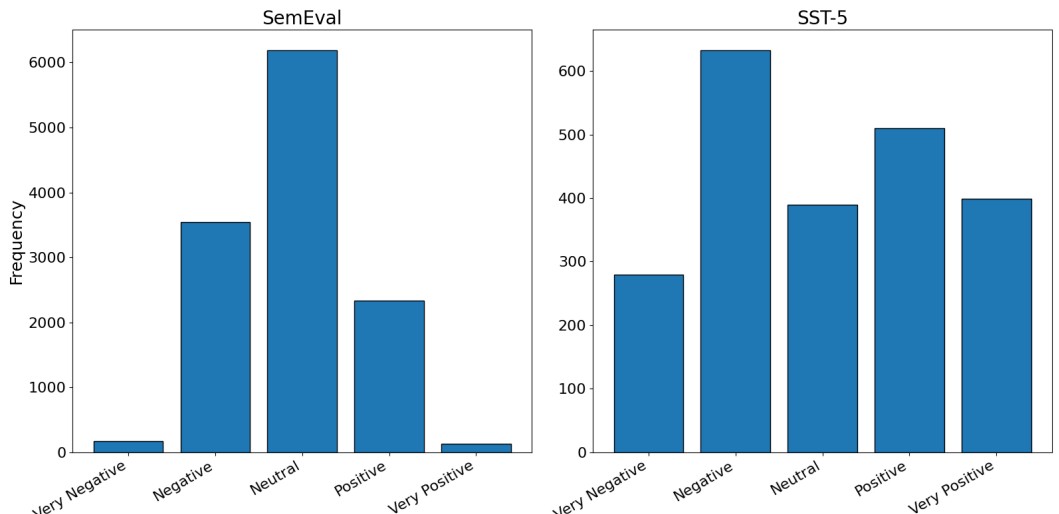

Figure 1: Label distribution on SemEval and SST-5 datasets.

In contrast, the Stanford Sentiment Treebank (SST-5) includes grammatically complete sentences from movie reviews. Thus, its primary purpose is to evaluate how well models understand the way sentiment is shaped by syntax, negation, contrastive conjunctions, sarcasm, and irony.

Since both datasets offer classification on a five-point scale, they provide an insightful diagnosis of a model's strengths and weaknesses, as they're based on two fundamentally different usages of language. Jointly, they provide a granular analysis of performance trade-offs at different model scales and costs.

The testing sample of SST-5 consists of 2210 instances, while that of SemEval consists of 12379 instances. Figure 1 shows that SemEval is significantly skewed towards "neutral" labels. This fact suggests the use of specific comparison metrics that consider such an imbalance, as shown in the next section.

Both datasets provide the text to be evaluated. However, SemEval goes a step further and offers the context for it. As a result, the prompts given to the models need to be different depending on the dataset from which the instance is drawn. The complete prompts are listed on sections A and B of the Appendix.

## 3.3 COMPARISON METRICS

We assess the performance of the models using the following metrics.

$$\text{Accuracy} = \frac{1}{n} \sum_{i=1}^{n} 1_{y_i = \hat{y}_i} \tag{1}$$

$$\text{Macro-Average Mean Absolute Error} = \frac{1}{5} \sum_{i=0}^{4} \frac{1}{|T_i|} \sum_{j \in T_i} |y_j - \hat{y}_j| \tag{2}$$

where $n$ represents the total amount of instances, $T_i$ the set of all instances with true label $i \in \{0, 1, 2, 3, 4\}$, $y_i$ and $\hat{y}_i$ represent, respectively the true and predicted label for the ith instance.

The framework first establishes a baseline with Accuracy, a conventional metric that measures the overall proportion of correct predictions. Its inclusion ensures comparability with a wide body of existing literature in sentiment polarity detection. However, Accuracy is notoriously misleading on ordinal datasets, since it treats all errors equally. For example, labeling a "very positive" text as "very negative" has the same weight as labeling it as "positive". Additionally, this metric may

misrepresent errors on imbalanced datasets, since it gives more weight to majority classes, which could mask poor performance on minority sentiments.

In order to address these problems, we also report the Macro-Average Mean Absolute Error. It is ideal for ordinal datasets, since it measures the average absolute difference between the predicted and true integer labels. Thus, slight errors are penalized more lightly than severe errors. Furthermore, this metric calculates the Mean Absolute Error individually for each class and takes the simple arithmetic mean across all classes, giving equal importance to each one, regardless of the number of instances. This makes it an essential tool for fairly assessing a model's performance on the imbalanced SemEval dataset.

These two metrics are then confronted with the "cost" aspect of the evaluation, assessed using the inverse of the Average End-to-End Latency, which captures the number of instances a given model answers per second.

## 4 RESULTS

All models were run on a single-GPU machine (NVIDIA RTX A5500, 24GB VRAM). A time limit of 100 hours was employed on both datasets. If a model exceeded the time limit, it was immediately terminated and tagged with 'TLE' (Time Limit Exceeded) in the results tables, which are available in the Appendix. We note that only one model (phi4-reasoning_14b) exceeded the established time limit.

Additionally, an instance was discarded for a given model if, for that instance, the model's answer contained more than one valid integer or no valid integers. The first case often occurs when the model attempts to explain its answer even when explicitly instructed not to do so. The second case often occurs when it refuses to answer. This policy was adopted to facilitate the extraction of the model's predictions and increase confidence in the reported results. The number of skipped instances for each model can be seen in section D of the Appendix. However, we emphasize that all models ranked in the top 3 according to our evaluation measures had a skipped instance rate of much less than 1%, and most of them had no skipped instances at all.

We ran the experiments for each model and calculated the Accuracy, the Macro-Average Mean Absolute Error, and the instances-per-second score. The results are presented in Figures 2 and 3. Figure 2 shows the instances-per-second score against Accuracy, while Figure 3 plots it against Macro-Average Mean Absolute Error for each evaluated model. Each model family is colored consistently.

We also highlight the models that determine the Pareto frontier on each graph. The Pareto frontier (or Pareto front) is the set of points that are not dominated by any other point. For any model in the Pareto frontier, no other model is better in one dimension without being worse in the other, showing the trade-off between two competing objectives, where improving one necessarily reduces the other.

For each figure, when available, the state-of-the-art performance is also depicted, allowing for a comparison of the current scene of open-weight LLMs with the best previous results for the corresponding metrics. In each figure, a small set of models performed significantly worse than the rest. This set was removed from the figure for visualization purposes. The full performance report can be found at section E of the Appendix.

We may notice that in the SemEval dataset, the current models, with no fine-tuning or special prompting techniques, achieve a new state-of-the-art performance for both Accuracy and Macro-Average Mean Absolute Error. Das & Pedersen (2024), yet unpublished, is our reference for the best Accuracy in the SemEval dataset. Their approach consists of using a fine-tuned BERT-based model. Four open-weight LLMs significantly overcome it, as shown in Figure 2, reaching the new best Accuracy value of 0.619 (previous best was 0.542).

Furthermore, as seen in Figure 3, for the metric of Macro-Average Mean Absolute Error, the best reported score, according to Rosenthal et al. (2017) was 0.481, which is surpassed by two open-weight LLMs, one of which (**gemma3_27b**) reached the value 0.462 (lower is better).

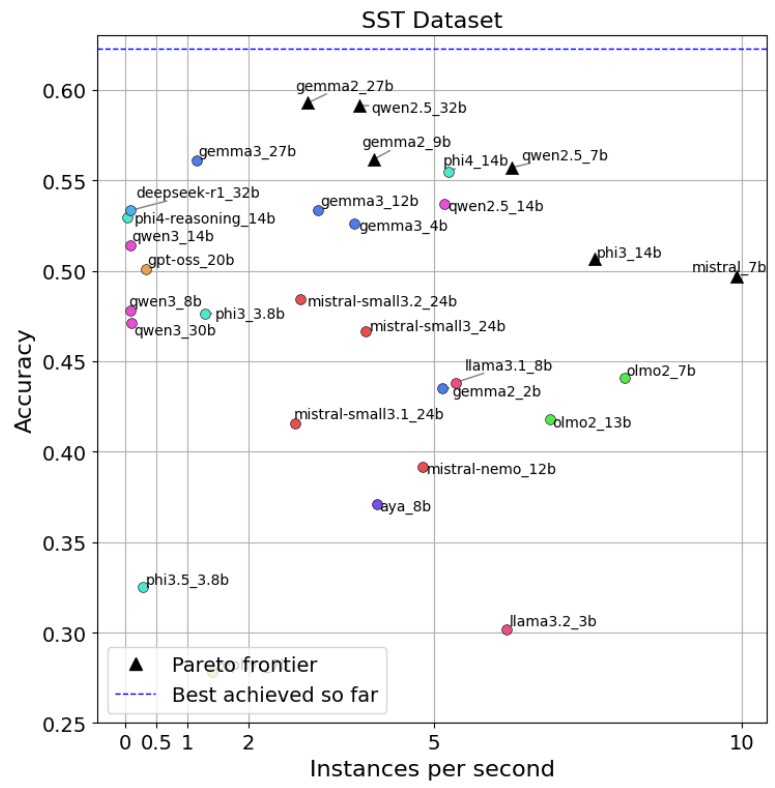

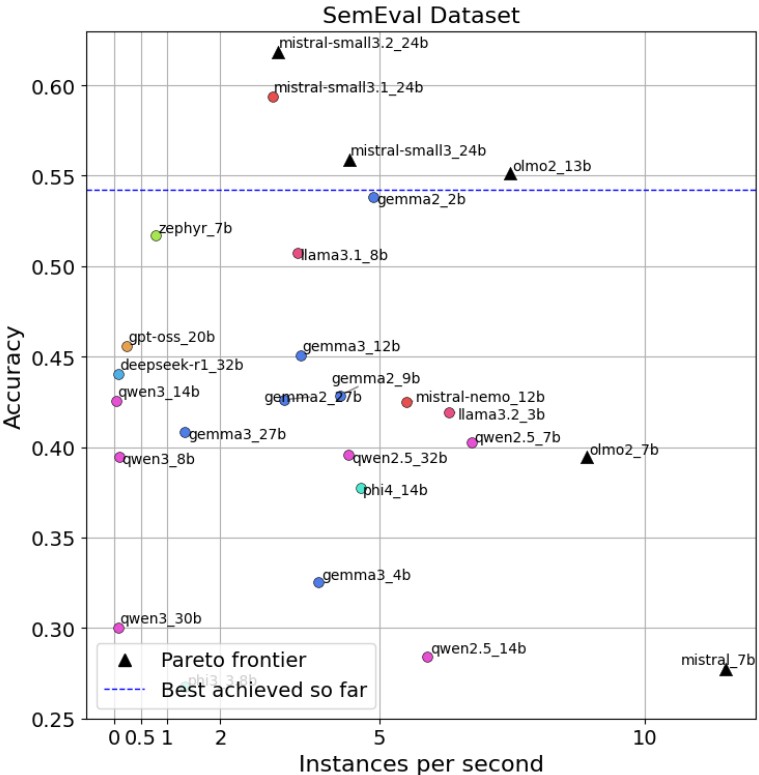

Figure 2: Accuracy and instances-per-second scores for each evaluated model.

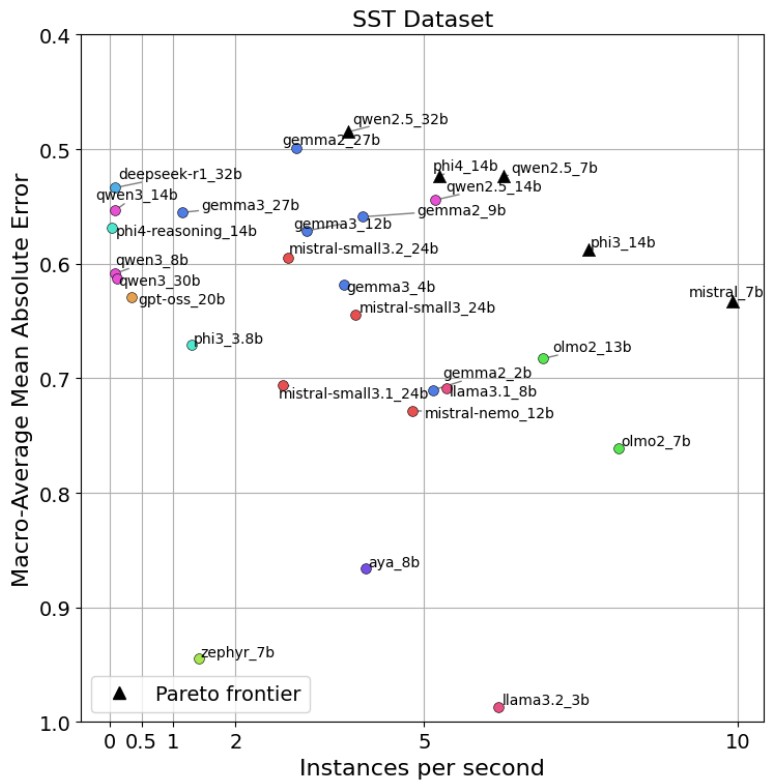

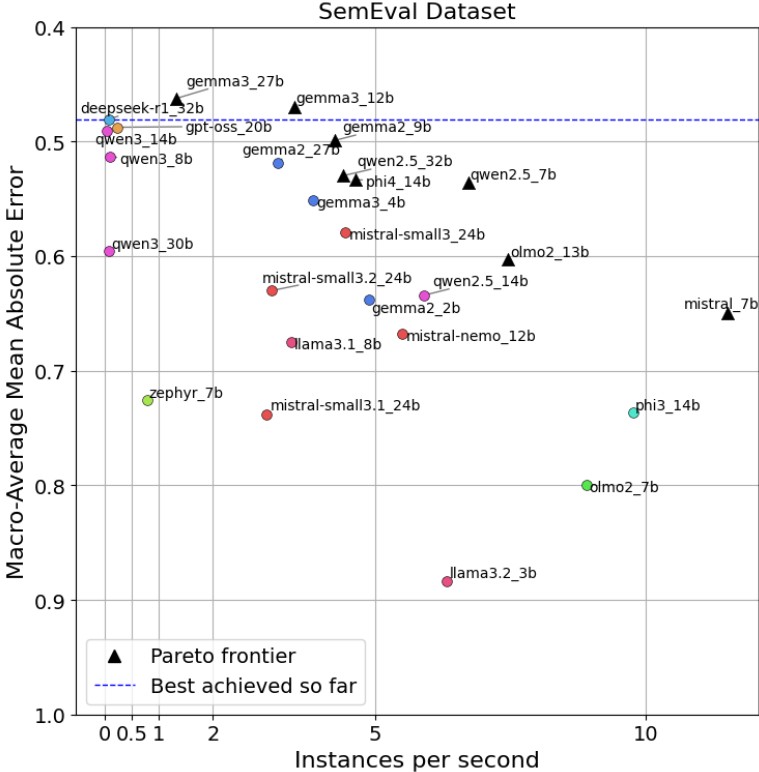

Figure 3: Macro-Average Mean Absolute Error and instances-per-second scores for each evaluated model.

## 5 CONCLUSION

This study presented a comprehensive benchmark of 31 open-weight large language models for multiclass sentiment polarity detection, focusing on the trade-off between performance and inference cost. Our zero-shot evaluation on the SemEval and SST-5 datasets reveals several key findings.

Most notably, we demonstrated that multiple contemporary open-weight LLMs, without any task-specific fine-tuning or elaborate prompting, can set a new state-of-the-art performance on the large-scale SemEval dataset in terms of both Accuracy (increasing performance from 0.542 to 0.619) and Macro-Average Mean Absolute Error (decreasing error from 0.481 to 0.462). On the smaller, compositionally complex SST-5 benchmark, these models achieve results that approach the existing state-of-the-art accuracy (0.5927 vs SOTA 0.6227).

The Pareto frontier analysis is an effective tool for identifying models that offer an optimal balance of performance and efficiency, providing a practical guide for selecting the most suitable model based on specific constraints, whether prioritizing maximum performance or high throughput. Our results confirm that the landscape of open-weight LLMs is mature enough to offer highly competitive, off-the-shelf solutions for nuanced NLP tasks, suggesting that specialized, fine-tuned models may not always be necessary for top-tier performance. Future work could extend this analysis by investigating other efficiency metrics, such as memory and energy consumption.

## REPRODUCIBILITY STATEMENT

We have made every effort to ensure the reproducibility of our results. The methodology, including model selection criteria, dataset descriptions, and evaluation metrics, is detailed in Section 3. The exact zero-shot prompts used for the SST-5 and SemEval datasets are provided in sections A and B of the Appendix. A complete list of the 31 evaluated models, including their origin and sizes, can be found in Section C of the Appendix, and we highlight that all experiments were conducted on the hardware specified in Section 4. Finally, a full report of each model's performance metrics is available in sections E and D of the Appendix, which allows for a complete reconstruction of our findings.

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

APPENDIX

## A  SST-5 PROMPT

Below, we describe the prompt structure used for the zero-shot classification task in the SST-5 dataset.

```
You are an expert sentiment analyst. Your task is to analyze the
sentiment polarity of the provided text.

Read the text carefully and classify its sentiment into one of
the following five categories.

The categories are defined as follows:
0: Very Negative - The text expresses a strong negative sentiment.
1: Negative - The text expresses a mild or weak negative sentiment.
2: Neutral - The text is objective, contains balanced sentiment,
or expresses no sentiment.
3: Positive - The text expresses a mild or weak positive sentiment.
4: Very Positive - The text expresses a strong positive sentiment.

You must respond with a single integer from 0, 1, 2, 3, or 4 and
nothing else.

Text to analyze:
```

## B  SEMEVAL PROMPT

Below, we describe the prompt structure used for the zero-shot classification task in the SemEval dataset.

```
You are an expert sentiment analyst. Your task is to analyze the
sentiment polarity of the provided text while taking into account
the given context (hashtag).

Read the text carefully and classify its sentiment into one of the
following five categories.

The categories are defined as follows:
0: Very Negative - The text expresses a strong negative sentiment.
1: Negative - The text expresses a mild or weak negative sentiment.
2: Neutral - The text is objective, contains balanced sentiment, or
expresses no sentiment.
3: Positive - The text expresses a mild or weak positive sentiment.
4: Very Positive - The text expresses a strong positive sentiment.

You must respond with a single integer from 0, 1, 2, 3, or 4 and
nothing else.

The first line contains the context (hashtag) and the second line
contains the text.
```

## C  LIST OF EVALUATED MODELS

Table 1: Complete list of large language models evaluated in this study.

| Model Name | Developer | Parameters (B) |
|---|---|---|
| aya | Cohere | 8 |
| deepseek-r1 | DeepSeek AI | 32 |
| gemma2 | Google DeepMind | 2, 9, 27 |
| gemma3 | Google DeepMind | 4, 12, 27 |
| gpt-oss | OpenAI | 20 |
| llama3.1 | Meta AI | 8 |
| llama3.2 | Meta AI | 3 |
| mistral | Mistral AI | 7 |
| mistral-nemo | Mistral AI | 12 |
| mistral-small3 | Mistral AI | 24 |
| mistral-small3.1 | Mistral AI | 24 |
| mistral-small3.2 | Mistral AI | 24 |
| olmo2 | Allen Institute for AI | 7, 13 |
| phi3 | Microsoft | 3.8, 14 |
| phi3.5 | Microsoft | 3.8 |
| phi4 | Microsoft | 14 |
| phi4-mini | Microsoft | 3.8 |
| phi4-reasoning | Microsoft | 14 |
| qwen2.5 | Alibaba Cloud | 7, 14, 32 |
| qwen3 | Alibaba Cloud | 8, 14, 30 |
| zephyr | Hugging Face H4 | 7 |

## D SKIPPED INSTANCES

Table 2: Number of instances with invalid output for each model and dataset. The output is invalid when it doesn't contain an integer in the specified range [0, 4] or when it contains multiple integers that satisfy the above criterion.

| Model | Dataset | Count | | Percentage (%) | |
|---|---|---|---|---|---|
| | | More than one number | No valid numbers | More than one number | No valid numbers |
| aya_8b | SemEval | 0 | 0 | 0.00 | 0.00 |
| | SST | 0 | 1 | 0.00 | 0.05 |
| deepseek-r1_32b | SemEval | 33 | 0 | 0.27 | 0.00 |
| | SST | 18 | 0 | 0.81 | 0.00 |
| gemma2_2b | SemEval | 0 | 0 | 0.00 | 0.00 |
| | SST | 0 | 0 | 0.00 | 0.00 |
| gemma2_9b | SemEval | 0 | 0 | 0.00 | 0.00 |
| | SST | 0 | 0 | 0.00 | 0.00 |
| gemma2_27b | SemEval | 0 | 0 | 0.00 | 0.00 |
| | SST | 0 | 0 | 0.00 | 0.00 |
| gemma3_4b | SemEval | 0 | 1 | 0.00 | 0.01 |
| | SST | 0 | 0 | 0.00 | 0.00 |

**Table 2 – continued from previous page**

| Model | Dataset | Count | | Percentage (%) | |
|---|---|---|---|---|---|
| | | More than one number | No valid numbers | More than one number | No valid numbers |
| gemma3_12b | SemEval | 0 | 0 | 0.00 | 0.00 |
| | SST | 0 | 0 | 0.00 | 0.00 |
| gemma3_27b | SemEval | 0 | 0 | 0.00 | 0.00 |
| | SST | 0 | 0 | 0.00 | 0.00 |
| gpt-oss_20b | SemEval | 2 | 8 | 0.02 | 0.06 |
| | SST | 0 | 0 | 0.00 | 0.00 |
| llama3.1_8b | SemEval | 333 | 126 | 2.69 | 1.02 |
| | SST | 2 | 1 | 0.09 | 0.05 |
| llama3.2_3b | SemEval | 0 | 114 | 0.00 | 0.92 |
| | SST | 0 | 0 | 0.00 | 0.00 |
| mistral-nemo_12b | SemEval | 0 | 0 | 0.00 | 0.00 |
| | SST | 0 | 0 | 0.00 | 0.00 |
| mistral-small3_24b | SemEval | 0 | 51 | 0.00 | 0.41 |
| | SST | 0 | 0 | 0.00 | 0.00 |
| mistral-small3.1_24b | SemEval | 8 | 21 | 0.06 | 0.17 |
| | SST | 0 | 0 | 0.00 | 0.00 |
| mistral-small3.2_24b | SemEval | 0 | 1 | 0.00 | 0.01 |
| | SST | 0 | 0 | 0.00 | 0.00 |
| mistral_7b | SemEval | 15 | 0 | 0.12 | 0.00 |
| | SST | 0 | 0 | 0.00 | 0.00 |
| olmo2_7b | SemEval | 0 | 53 | 0.00 | 0.43 |
| | SST | 0 | 0 | 0.00 | 0.00 |
| olmo2_13b | SemEval | 2 | 35 | 0.02 | 0.28 |
| | SST | 0 | 0 | 0.00 | 0.00 |
| phi3.5_3.8b | SemEval | 3620 | 5222 | 29.24 | 42.18 |
| | SST | 639 | 737 | 28.91 | 33.35 |
| phi3_3.8b | SemEval | 92 | 11378 | 0.74 | 91.91 |
| | SST | 101 | 268 | 4.57 | 12.13 |
| phi3_14b | SemEval | 0 | 0 | 0.00 | 0.00 |
| | SST | 3 | 1 | 0.14 | 0.05 |
| phi4-mini_3.8b | SemEval | 10555 | 548 | 85.27 | 4.43 |
| | SST | 1917 | 83 | 86.74 | 3.76 |
| phi4-reasoning_14b | SemEval | TLE | TLE | TLE | TLE |
| | SST | 8 | 0 | 0.36 | 0.00 |
| phi4_14b | SemEval | 14 | 0 | 0.11 | 0.00 |
| | SST | 0 | 0 | 0.00 | 0.00 |
| qwen2.5_7b | SemEval | 0 | 0 | 0.00 | 0.00 |
| | SST | 0 | 0 | 0.00 | 0.00 |
| qwen2.5_14b | SemEval | 0 | 0 | 0.00 | 0.00 |
| | SST | 0 | 0 | 0.00 | 0.00 |
| qwen2.5_32b | SemEval | 0 | 0 | 0.00 | 0.00 |
| | SST | 0 | 0 | 0.00 | 0.00 |
| qwen3_8b | SemEval | 4 | 4 | 0.03 | 0.03 |

**Table 2 – continued from previous page**

| Model | Dataset | Count | | Percentage (%) | |
|---|---|---|---|---|---|
| | | More than one number | No valid numbers | More than one number | No valid numbers |
| | SST | 0 | 2 | 0.00 | 0.09 |
| qwen3_14b | SemEval | 19 | 12 | 0.15 | 0.10 |
| | SST | 0 | 0 | 0.00 | 0.00 |
| qwen3_30b | SemEval | 27 | 0 | 0.22 | 0.00 |
| | SST | 2 | 0 | 0.09 | 0.00 |
| zephyr_7b | SemEval | 5175 | 0 | 41.80 | 0.00 |
| | SST | 758 | 0 | 34.30 | 0.00 |

# E   FULL PERFORMANCE REPORT

Table 3: Model Performance Comparison on SST and SemEval Datasets.
Models in bold appear in at least one of the Pareto frontier analyses.

| Model | Dataset | Accuracy | Macro-Average MAE | Instances per second |
|---|---|---|---|---|
| **gemma2_9b** | SST | 0.5615 | 0.5590 | 4.0293 |
| | SemEval | 0.4284 | 0.4990 | 4.2545 |
| **gemma2_27b** | SST | 0.5927 | 0.4992 | 2.9586 |
| | SemEval | 0.4262 | 0.5186 | 3.2016 |
| **gemma3_12b** | SST | 0.5334 | 0.5713 | 3.1319 |
| | SemEval | 0.4509 | 0.4701 | 3.5117 |
| **gemma3_27b** | SST | 0.5610 | 0.5552 | 1.1552 |
| | SemEval | 0.4085 | 0.4622 | 1.3271 |
| **mistral_7b** | SST | 0.4968 | 0.6329 | 9.9169 |
| | SemEval | 0.2772 | 0.6496 | 11.5224 |
| **mistral-small3_24b** | SST | 0.4665 | 0.6444 | 3.9012 |
| | SemEval | 0.5588 | 0.5794 | 4.4383 |
| **mistral-small3.2_24b** | SST | 0.4841 | 0.5949 | 2.8309 |
| | SemEval | 0.6185 | 0.6301 | 3.0789 |
| **olmo2_7b** | SST | 0.4407 | 0.7612 | 8.0955 |
| | SemEval | 0.3945 | 0.7992 | 8.9063 |
| **olmo2_13b** | SST | 0.4176 | 0.6827 | 6.8970 |
| | SemEval | 0.5516 | 0.6022 | 7.4590 |
| **phi3_14b** | SST | 0.5063 | 0.5875 | 7.6151 |
| | SemEval | 0.2234 | 0.7363 | 9.7682 |
| **phi4_14b** | SST | 0.5547 | 0.5234 | 5.2380 |
| | SemEval | 0.3776 | 0.5332 | 4.6420 |
| **qwen2.5_7b** | SST | 0.5570 | 0.5235 | 6.2647 |
| | SemEval | 0.4025 | 0.5361 | 6.7326 |
| **qwen2.5_32b** | SST | 0.5914 | 0.4849 | 3.7918 |
| | SemEval | 0.3955 | 0.5297 | 4.4131 |
| aya_8b | SST | 0.3712 | 0.8662 | 4.0777 |
| | SemEval | 0.2064 | 1.3947 | 4.3348 |

**Table 3 – continued from previous page**

| Model Name | Dataset | Accuracy | Macro-Average MAE | Instances per second |
|---|---|---|---|---|
| deepseek-r1_32b | SST | 0.5333 | 0.5337 | 0.0797 |
| | SemEval | 0.4403 | 0.4805 | 0.0788 |
| gemma2_2b | SST | 0.4352 | 0.7102 | 5.1468 |
| | SemEval | 0.5381 | 0.6375 | 4.8875 |
| gemma3_4b | SST | 0.5262 | 0.6184 | 3.7162 |
| | SemEval | 0.3251 | 0.5510 | 3.8505 |
| gpt-oss_20b | SST | 0.5009 | 0.6293 | 0.3372 |
| | SemEval | 0.4559 | 0.4875 | 0.2315 |
| llama3.1_8b | SST | 0.4381 | 0.7089 | 5.3542 |
| | SemEval | 0.5075 | 0.6749 | 3.4502 |
| llama3.2_3b | SST | 0.3018 | 0.9872 | 6.1872 |
| | SemEval | 0.4192 | 0.8834 | 6.3181 |
| mistral-nemo_12b | SST | 0.3918 | 0.7283 | 4.8196 |
| | SemEval | 0.4251 | 0.6680 | 5.5031 |
| mistral-small3.1_24b | SST | 0.4158 | 0.7060 | 2.7557 |
| | SemEval | 0.5938 | 0.7383 | 2.9842 |
| phi3_3.8b | SST | 0.4763 | 0.6703 | 1.2966 |
| | SemEval | 0.2673 | 1.5961 | 1.3347 |
| phi3.5_3.8b | SST | 0.3249 | 1.2286 | 0.2751 |
| | SemEval | 0.2010 | 1.6464 | 0.5777 |
| phi4-mini_3.8b | SST | 0.1904 | 1.5974 | 0.1877 |
| | SemEval | 0.2296 | 1.5598 | 0.2325 |
| phi4-reasoning_14b | SST | 0.5295 | 0.5690 | 0.0325 |
| | SemEval | TLE | TLE | TLE |
| qwen2.5_14b | SST | 0.5371 | 0.5443 | 5.1815 |
| | SemEval | 0.2840 | 0.6342 | 5.8950 |
| qwen3_8b | SST | 0.4778 | 0.6087 | 0.0749 |
| | SemEval | 0.3946 | 0.5134 | 0.0989 |
| qwen3_14b | SST | 0.5140 | 0.5533 | 0.0840 |
| | SemEval | 0.4256 | 0.4904 | 0.0417 |
| qwen3_30b | SST | 0.4710 | 0.6127 | 0.1046 |
| | SemEval | 0.3002 | 0.5953 | 0.0795 |
| zephyr_7b | SST | 0.2782 | 0.9449 | 1.4090 |
| | SemEval | 0.5172 | 0.7253 | 0.7811 |

