# OpenReview forum: "Large Language Models outperform state-of-the-art methods on multiclass sentiment polarity detection"
_ICLR.cc/2026/Conference — Submitted to ICLR 2026_

### Official Review · Reviewer_HQCw · 2025-10-27

**Soundness:** 2
**Presentation:** 1
**Contribution:** 1
**Rating:** 2
**Confidence:** 4

**Summary:**

The paper systematically benchmarks 31 open-weight LLMs on five-class sentiment classification (SemEval-2017 Task 4C and SST-5).
Using a zero-shot setup and measuring accuracy, macro-average mean absolute error, and instances-per-second, it finds that several open models (e.g., Gemma-3 27B, Mistral-small3.2 24B) exceed previous SOTA on SemEval without fine-tuning. The paper also provides a Pareto frontier to illustrate trade-offs between accuracy and inference speed.

**Strengths:**

1. Evaluates a large, diverse set of open-weight LLMs across families and sizes, offering a valuable comparative resource for the community.
2. Goes beyond accuracy by including Macro-Average MAE and instances-per-second, capturing both quality and efficiency trade-offs.

**Weaknesses:**

1. The study mainly reuses existing datasets and evaluation setups, focusing on performance comparison rather than introducing any new modeling idea, benchmark design, or evaluation framework.

2. The introduction section references only a small portion of prior work, offering limited connection to the broader sentiment analysis and LLM evaluation literature.

3. The experiments rely on a single fixed prompt for each dataset, without examining how small changes in wording might affect model behavior, which is a key factor in large language model reliability.

4. The paper does not address potential overlap between the test data and model pretraining corpora, which is particularly relevant because both evaluated datasets are long-established and widely circulated.

5. The empirical analysis remains narrow: only two datasets are tested, and there is little exploration of error patterns or qualitative reasoning behind the models’ successes and failures.

**Questions:**

NA

---

### Official Review · Reviewer_Xx3W · 2025-11-01

**Soundness:** 2
**Presentation:** 2
**Contribution:** 2
**Rating:** 2
**Confidence:** 5

**Summary:**

This paper evaluate the performance of 31 open-source large language models (open-weight LLMs) on the task of multiclass sentiment polarity detection, whilst examining the trade-off between performance and inference cost.

**Strengths:**

Tables and graphs provide a more straightforward presentation. The appendix contains a comprehensive list of models, prompts, failure sample rates, and detailed results.

**Weaknesses:**

1.Unclear motivation and innovation
The paper does not propose novel algorithms or provide clear motivation demonstrating the necessity of this work. It reads more like a report expressing a single thesis: that untuned large models achieve superior sentiment classification performance. However, this falls short of the requirements for an academic paper.
2.Writing and structural issues
The paper's structure is rather loose, and its language is more akin to that of a report or technical white paper;

**Questions:**

1.Sentiment classification is not a particularly challenging task. What is the point of evaluating large language models on this task?
2.The paper claims that the large model's performance surpasses state-of-the-art models, yet this is not explicitly demonstrated within the paper, such as through tables or images.

---

### Official Review · Reviewer_kCuQ · 2025-11-06

**Soundness:** 3
**Presentation:** 1
**Contribution:** 1
**Rating:** 0
**Confidence:** 4

**Summary:**

The paper evaluates the performance and efficiency of 31 open-weight LLMs on two sentiment analysis tasks, providing comprehensive results. This is the paper’s only contribution.

**Strengths:**

- A large number (31) of open-weight LLMs are evaluated on a multi-label sentiment analysis dataset, with both accuracy and efficiency metrics reported.

**Weaknesses:**

- The paper’s contribution is minimal. It only provides a joint evaluation of performance and efficiency of LLMs on sentiment analysis tasks. The so-called Pareto frontier analysis, while practical and informative, is not a novel evaluation approach.

- Although the reported results may be useful to the community, they are more appropriate for an initial technical report or blog post. In contrast, conferences such as ICLR emphasize novelty in research questions or methodologies—introducing new ideas, not merely new results.

- The paper’s content is rather superficial. Of the 8 pages of main text, less than half a page is used to present the benchmark label distribution (Figure 1), and two full pages are devoted to displaying largely redundant evaluation results (Figures 2 & 3). Moreover, about half a page remains blank. Given the level of detail, the same content could likely be presented in 4–5 pages through concise writing.

- The reference list is short, containing only 13 works, with just one citation in the introduction. This suggests a limited understanding of prior research in the field. Also, the presentation is quite immature, with many informal, repetitive, and unprofessional expressions throughout the paper.

- To make the work suitable for a conference submission, the authors could consider extending beyond simple evaluation and explore potential research contributions, such as:
   - If focusing on the accuracy–efficiency trade-off, propose methods to optimize this trade-off—e.g., achieving higher accuracy with smaller or more efficient LLMs.
   - If focusing on evaluation methodology, investigate the limitations of existing evaluation practices—are they efficient, and do they truly reflect model capabilities?
   - Alternatively, broaden the research scope beyond sentiment polarity detection to other, more general NLP tasks.
   - ...

**Questions:**

no questions

---

### Meta-Review · Area_Chair_MrbB · 2026-01-12

**Summary:**

This paper evaluates a diverse set of 31 open-weight LLMs across families and sizes, on sentiment analysis tasks. Unfortunately, the papers contributions are unanimously considered minimal due to lack of novelty, unclear motivation, innovation and writing quality below ICLR standards. The paper comes across as a preliminary report with minimal references, instead of a complete submission.

**Reviewer Concerns:**

NA - No rebuttal response.

**Reviewer Scores:**

NA

---

### Decision · Program_Chairs · 2026-01-26

Reject